# Kinetic resolution of cyclic benzylic azides enabled by site- and enantioselective C(sp$^3$)–H oxidation

Pengbo Ye[1,2], Aili Feng[1,2], Lin Wang[1,2], Min Cao[1], Rongxiu Zhu[1] & Lei Liu [1✉]

Catalytic nonenzymatic kinetic resolution (KR) of racemates remains one of the most powerful tools to prepare enantiopure compounds, which dominantly relies on the manipulation of reactive functional groups. Moreover, catalytic KR of organic azides represents a formidable challenge due to the small size and instability of the azido group. Here, an effective KR of cyclic benzylic azides through site- and enantioselective C(sp$^3$)–H oxidation is described. The manganese catalyzed oxidative KR reaction exhibits good functional group tolerance, and is applicable to a range of tetrahydroquinoline- and indoline-based organic azides with excellent site- and enantio-discrimination. Computational studies elucidate that the effective chiral recognition is derived from hydrogen bonding interaction between substrate and catalyst.

[1] School of Chemistry and Chemical Engineering, Shandong University, Jinan 250100, China. [2] These authors contributed equally: Pengbo Ye, Aili Feng, Lin Wang. ✉email: leiliu@sdu.edu.cn

Catalytic nonenzymatic kinetic resolution (KR) of racemates is one of the most powerful and practical tools to prepare valuable enantiopure targets, especially in cases where other methods are not possible or provide insufficient enantiocontrol[1–5]. Chiral organic azides are versatile synthetic precursors for a range of nitrogen-containing molecules and have found dramatically expanded utility in medicine, biology, and material science[6–15]. However, catalytic KR to provide optically pure azides has remained elusive, principally due to two essential features of the azido moiety: (1) the instability hampering the design of new reactivity with excellent chemoselectivity; (2) the small size hampering the achievement of effective chiral recognition[16–21]. Existing isolated examples always focused on manipulating the azido moiety through azide–alkyne cycloaddition (AAC)[22–26] (Fig. 1A) or extra reactive functional groups preinstalled in substrates (Fig. 1B)[27,28], which typically suffer from the use of excess azide substrates, poor chiral recognition, and narrow substrate scope. Developing an effective KR of organic azides relying on the reactivity of C(sp³)–H bonds would be highly desired[29].

Nonenzymatic site- and enantioselective oxidation of ubiquitous C(sp³)–H bonds with a general scope and predictable selectivity represents a paradigm shift in the standard logic of organic synthesis[30,31]. However, such research topic has remained

a formidable challenge, and current studies typically suffer from moderate enantioselectivity, low substrate conversion, and narrow substrate scope[32–44]. In particular, catalytic KR through C(sp³)–H oxidation dominantly focused on secondary alcohols[45–52] and amines[53,54] due to their high and well-known oxidized reactivity together with the presence of a strong interaction site with catalyst for efficient chiral recognition[55,56]. To our knowledge, selective oxidation of C(sp³)−H bond adjacent to azido moiety remains elusive. Moreover, organic azide lacks such an effective interaction site to direct substrate to an ideal location in the transition state. Therefore, chiral recognition of chemically similar C(sp³)−H bonds adjacent to azido group of two enantiomers would be difficult to accomplish.

Herein, we report the KR of organic azides through site- and enantioselective C(sp³)–H bond oxidation (Fig. 1C). First, given the significance of benzo-fused nitrogen-containing heterocycles in modern pharmacology, we choose a range of racemic benzylic azides bearing such skeletons as substrates. Second, we select the readily modifiable salen as the basal ligand to search for suitable base-metal catalyst. Third, varying the protecting group on the nitrogen moiety might also provide an opportunity to tune the chiral recognition. Based on these considerations, a range (36 examples) of cyclic benzylic azides participate in oxidative KR with good to excellent selectivity factors (s up to 95).

**A. KR of benzylic azides through azide-alkyne cycloaddition (AAC) (ref. 22 and 23)**

- relying on the reactivity of azido group
- poor chiral recognition (s < 8)
- excess azide substrate (2.5 equiv)

**B. KR of allylic azides through asymmetric dihydroxylation of alkene (ref. 27)**

- relying on the reactivity of alkene

**C. KR of cyclic benzylic azides through asymmetric C(sp³)–H oxidation (this work)**

challenges
- elusive oxidized reactivity
- lacking an effective interaction site
- competing site-selectivity ($C_4$-H vs $C_2$-H)

advantages
- relying on the reactivity of C-H bond
- effective chiral recognition (s up to 95)
- excellent site-selectivity
- azide substrate as limiting agent

**Fig. 1 Overview of KR methods to prepare chiral organic azides. A** KR of benzylic azides through azide–alkyne cycloaddition. **B** KR of allylic azides through asymmetric dihydroxylation of alkene. **C** KR of cyclic benzylic azides through asymmetric C(sp³)−H oxidation.

**Table 1 Reaction condition optimization[a].**

| Entry | Catalyst | Conv. (%)[b] | ee (%)[c] | s[d] |
|---|---|---|---|---|
| 1 | C1 or C2 | <5 | n.d. | n.d. |
| 2 | C3 | 60 | 25 | 1.7 |
| 3 | C4 | 57 | 30 | 2.1 |
| 4 | C5 | 56 | 36 | 2.5 |
| 5 | C6 | 60 | 39 | 2.4 |
| 6 | C7 | 55 | 45 | 3.3 |
| 7 | C8 | 50 | 34 | 2.8 |
| 8 | C9 | 49 | 50 | 5.1 |
| 9 | C10 | 52 | 70 | 9.5 |
| 10 | C11 | 49 | 79 | 25 |
| 11 | C12 | 50 | 86 | 37 |
| 12[e] | C12 | <20 | n.d. | n.d. |
| 13[f] | C12 | 52 | 94 | 50 |
| 14[g] | C12 | 52 | 98 | 91 |

n.d. not determined.

[a]Reaction condition: to rac-**1a** (0.1 mmol) and catalyst (5 mol%) in EtOAc (1.0 ml) at rt was added PhIO (0.08 mmol) as two portions in 2 h intervals, unless otherwise noted.

[b]Conversion was calculated from yield of recovered **1a**.

[c]Determined by chiral HPLC analysis.

[d]Selectivity (s) values were calculated through the equation $s = \ln[(1 - C)(1 - ee)]/\ln[(1 - C)(1 + ee)]$, where C is the conversion.

[e]NaClO or 30% aqueous $H_2O_2$ as oxidant.

[f]PhIO was added as four portions in 1 h intervals over 3 h.

[g]PhIO was added as eight portions in 0.5 h intervals over 3.5 h.

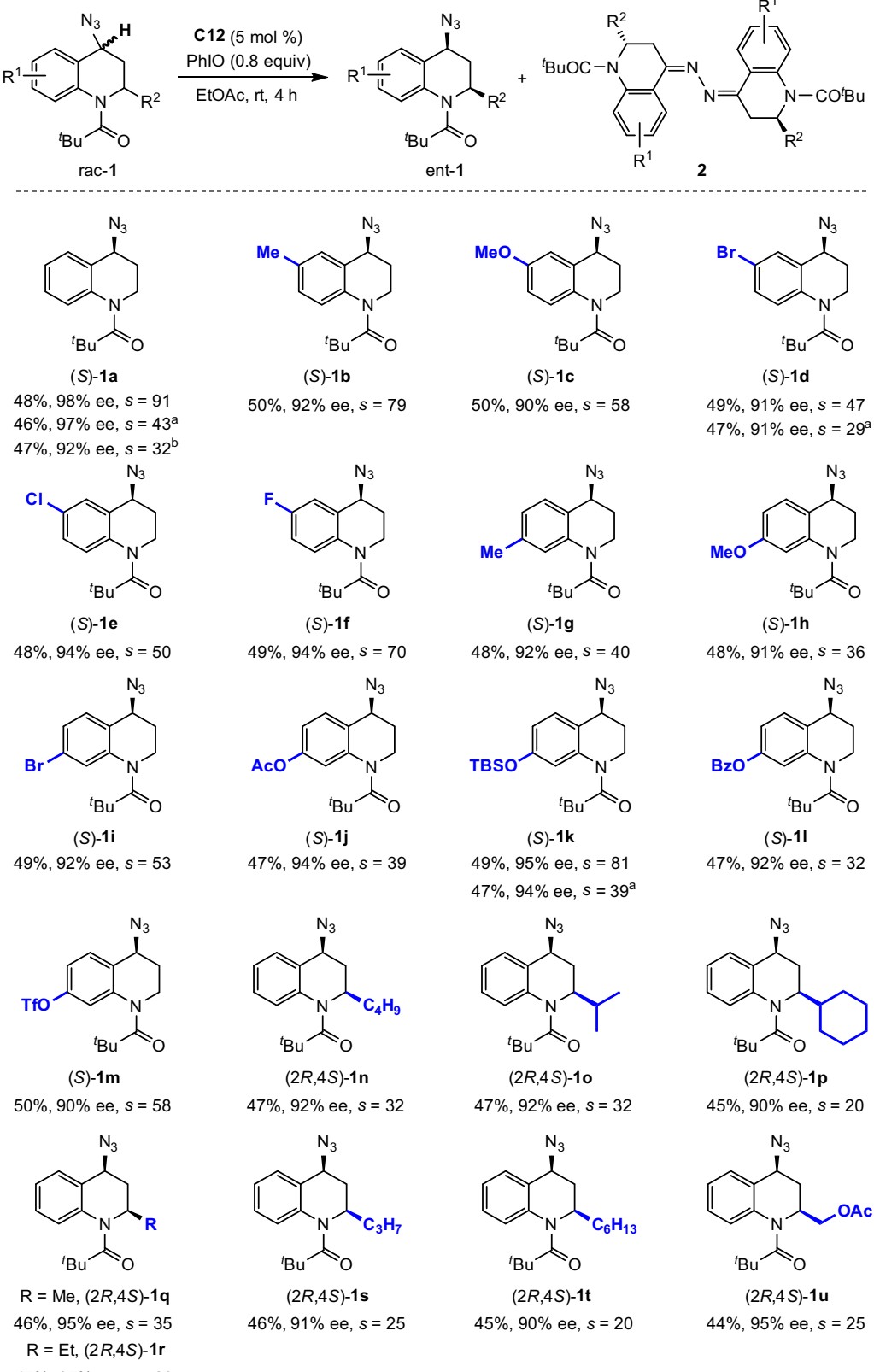

**Fig. 2 Kinetic resolution of THQ-based organic azides.** Conditions: rac-**1** (0.1 mmol), PhIO (0.08 mmol, addition as 8 portions in 30-min intervals over 3.5 h), and **C12** (5 mol%) in ethyl acetate (1.0 ml) at rt for 4 h. [a]Reaction with 0.5 mmol rac-**1**. [b]Reaction with 1.0 gram of rac-**1**.

## Results and discussion

**Reaction condition optimization.** The oxidative KR of tetra-hydroquinoline (THQ) based organic azide rac-**1a** was selected as the model reaction for optimization (Table 1). In the presence of PhIO as the oxidant, no reaction was observed for either chiral Fe(salen) **C1** or Co(salen) **C2** (entry 1). Chiral Mn(salen) **C3** exhibited good oxidation catalysis reactivity, though poor chiral recognition was obtained (entry 2). Oxidation proceeded with excellent site selectivity at the $C_4$–H bond adjacent to azido moiety over $C_2$–H bond α to amide motif, affording azine **2a** as oxidized product. Mn(salen) **C4** having cyclohexanediamine skeleton provided better results than **C3** with 1,2-diphenyl-1,2-ethanediamine (entry 3). Careful examination of the substituent effects on the basal salen ligand revealed **C12** with 2,4-difluor-ophenyl moieties at C3(3') sites to be optimal (entries 4–10). Other oxidants such as $H_2O_2$ and NaClO afforded inferior oxidation reactivity (entry 11). Addition of PhIO as eight equal portions in 30 min intervals was beneficial for achieving an extremely high level of chiral discrimination, and (S)-**1a** was isolated in 48% yield with 98% ee (s = 91, entries 12 and 13).

**Substrate scope.** The scope of oxidative KR of THQ-based organic azides was explored (Fig. 2). In general, both electron-rich and -deficient THQ skeletons were well tolerated, as demonstrated by effective access to optically pure **1a**-**1m** with good to excellent selectivity factors (s = 32–91). Resolution efficiency was not impaired for reaction on a 0.5 mmol scale. Common functional groups, including halide, acetate, silyl ether, benzoate, and triflate, were tolerated for further manipulation. Racemic THQ-based organic azides bearing two stereocenters were also suitable components with good enantio-discrimination. Oxidative KR of cis-2,4-disubstituted rac-**1n** proceeded, furnishing (2R,4S)-**1n** in 47% yield with 92% ee (The absolute configuration of recovered **1n** was determined by X-ray diffraction analysis. See the Supporting Information for details). The reaction was not sensitive to the steric hindrance of $C_2$-substituents, as demonstrated by access to respective enantiopure **1o**-**1u** with good selectivity factors (s = 20–35).

Indoline represents the other type of biologically important benzo-fused nitrogen-containing heterocycle. Accordingly, the applicability of the oxidative KR strategy in enantioselective

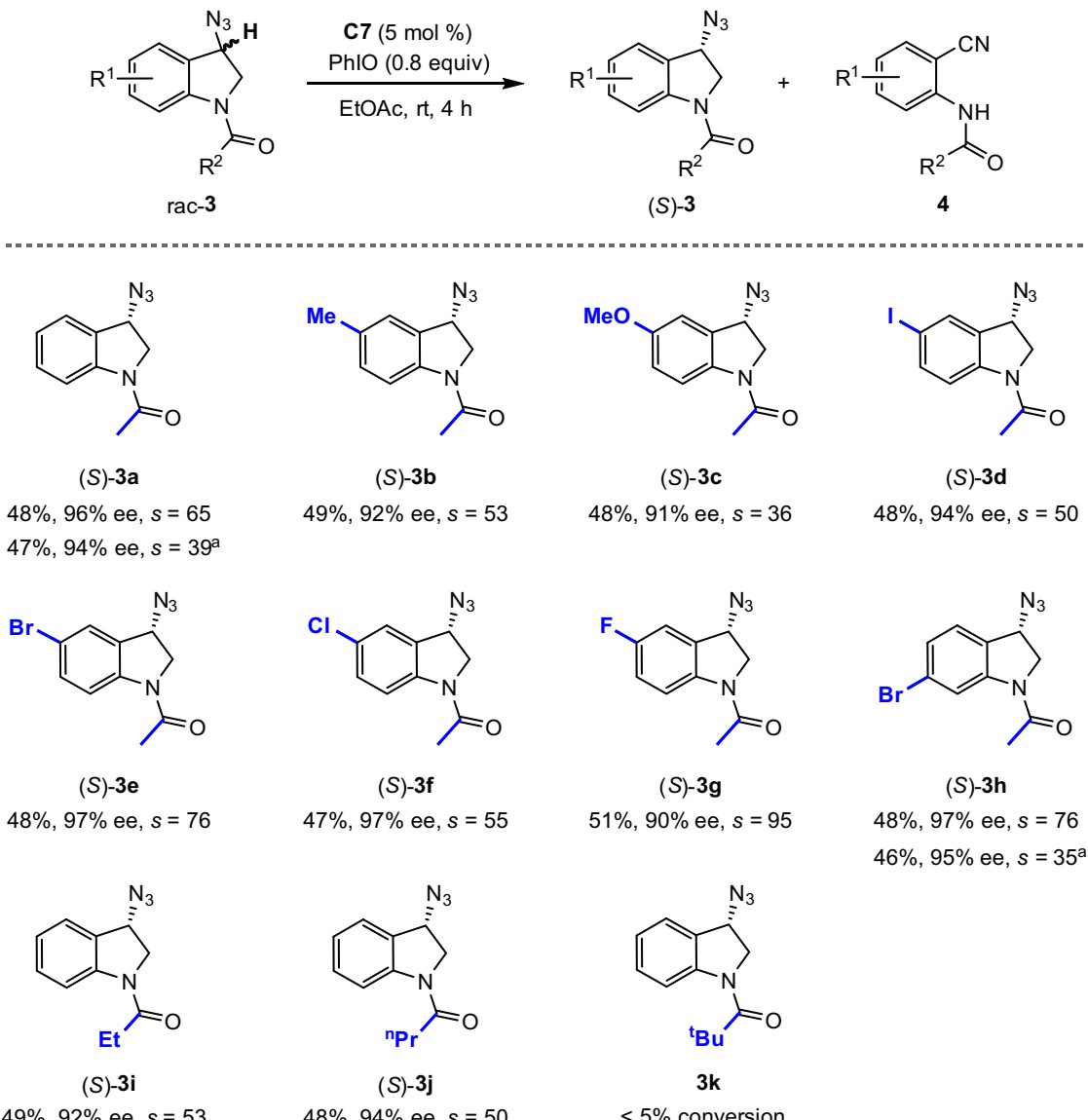

**Fig. 3 Kinetic resolution of indoline-based organic azides.** Conditions: rac-**3** (0.1 mmol), PhIO (0.08 mmol, addition as 8 portions in 30-min intervals over 3.5 h), and **C7** (5 mol%) in ethyl acetate (1.0 ml) at rt for 4 h. [a]Reaction with 0.5 mmol rac-**3**.

access to indoline-based azides was next explored (Fig. 3). When Mn(salen) **C7** was used as catalyst, oxidative KR of cyclic benzylic azide rac-**3a** proceeded, furnishing 2-aminobenzonitrile **4a** as oxidized product together with recovered (S)-**3a** in 48% yield with 96% ee (s = 65) (The absolute configuration of recovered **3a** was determined by X-ray diffraction analysis. See the Supporting Information for details). Notably, the reaction exhibited excellent site selectivity at the $C_3$–H bond adjacent to azido moiety over $C_2$–H bond α to amide motif. Indolines rac-**3b**-**3h** bearing electronically varied groups around the arene moiety were tolerated with a high level of chiral recognition (s = 36–95). Selectivities were not impaired for reaction on a 0.5 mmol scale. Substrates bearing other N-acyl groups, such as propanoyl (**3i**) and butyryl (**3j**), were also tolerated, though no reaction was observed for pivaloyl one (**3k**).

**Synthetic applications.** Manipulating the azido moiety through copper-catalyzed AAC would allow facile integration of the other biologically important molecules into N-heterocycles for drug discovery. For example, vitamin E derivative (**5**) with potent antioxidation activity and estrone derivative (**7**) for treating abnormalities associated with menopause are efficiently installed into THQ skeleton using triazole as a linker, respectively (Fig. 4).

**Mechanistic and DFT studies.** Control experiments were conducted to get a preliminary understanding of the reaction mechanism (Fig. 5). The relationship between ee values of Mn(salen) catalyst and recovered substrate was explored, showing that the enantioselectivity of recovered **3a** is proportional to the ee of **C7** (Fig. 5A). The absence of nonlinear effects indicated that the reaction might not involve heterochiral aglomerates[57,58]. No reaction was observed for stoichiometric Mn(salen) **C12** mediated reaction in the absence of PhIO, suggesting that oxoMn(V) might be the species in charge of C–H oxidation (Fig. 5B). A competition deuterium kinetic isotope

effect (KIE) study, using a mixture of rac-**1a** and [D]-rac-**1a**, revealed a KIE of 2.7 (Fig. 5C). The observation implied that C−H bond cleavage might be involved in the rate-determining step. The substituent effect of different acyl groups on THQ-based azides was explored (Fig. 5D). Several aspects of the data merit further comment. Firstly, no reaction was observed for N-acyl substituted **9a**. Secondly, the oxidized reactivity was gradually enhanced as the increasing numbers of methyl groups at α-position of the carbonyl moiety (**9a**-**9c** and **1a**). Thirdly, the oxidative reactivity was lost when placing an oxygen atom between the carbonyl and $^tBu$ groups. In general, more sterically hindered substrates should exhibit lower reactivity than that of less sterically hindered ones. We speculated that the opposite trend observed for THQ-based substrates in Fig. 5D might originate from the non-covalent interaction between the α-alkyl group of carbonyl moiety and Mn(salen) catalyst. The deuteration effect of the N-acyl moiety of indoline-based substrates was next evaluated (Fig. 5E). No oxidative conversion was observed for [D]-**3a**, indicating that the sp³ C−H bond at α-position of carbonyl motif is crucial to the reactivity of substrate **3a**.

According to the generally accepted mechanism of manganese-catalyzed C(sp³)−H oxidation and the control experiments, a plausible mechanistic pathway for oxidative KR of benzylic azides was suggested (Fig. 6)[59]. Chiral $Mn^{III}$ catalyst is first oxidized by PhIO affording $oxoMn^V$ intermediate. THQ rac-**1a** underwent hydrogen atom transfer (HAT) to $oxoMn^V$, giving benzylic radical **10** and $Mn^{IV}$−OH. Finally, azide **10** decomposed by losing molecular nitrogen to form iminyl radical **11**, which immediately dimerized to provide **2a**[60]. $Mn^{IV}$−OH dimerized by releasing $H_2O$ to generate μ-oxo bridged dimer $Mn^{IV}OMn^{IV}$, which might undergo disproportionation reaction regenerating $Mn^{III}$ precursor and $oxoMn^V$ species for the catalytic cycle[61–63]. Based on the absolute configuration of recovered THQs, (R)-**1a** should be oxidized more preferentially than (S)-**1a**. With respect to the oxidative KR of indoline-based benzylic azides, rac-**3a** might proceed through a similar HAT

**Fig. 4 Synthetic applications. A** $CuSO_4$-catalyzed AAC of optically pure THQ-based organic azide and vitamin E derivative. **B** CuI-catalyzed AAC of optically pure THQ-based organic azide and estrone derivative.

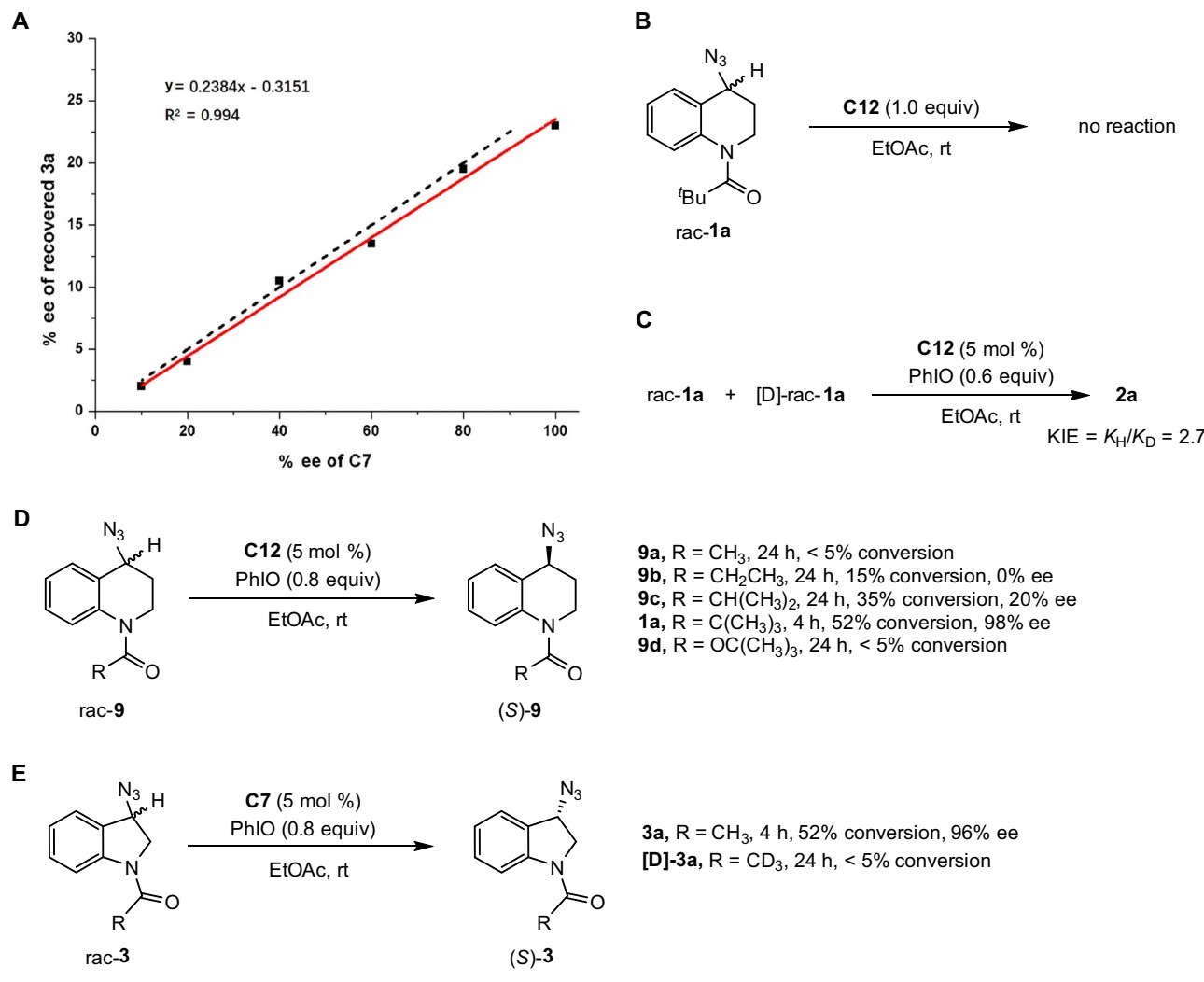

**Fig. 5 Control experiments. A** Plot of enantiomeric excess of recovered **3a** versus the enantiomeric excess of **C7** at 20% conversion. The dotted line symbolized the linear correlation. **B** Stoichiometric Mn(salen) **C12** mediated control experiment in the absence of PhIO. **C** The intermolecular kinetic isotope effect. **D** The *N*-acyl substituent effect for THQ-based azides. **E** Deuterated control experiment of indoline-based azide.

process to oxoMn$^V$ species producing benzylic radical **12**, which underwent azide collapse followed by C−C bond cleavage forming radical **14**. Alkyl radical **14** underwent oxygen rebound with Mn$^{IV}$−OH followed by hemiaminal decomposition, generating 2-aminobenzonitrile **4a** together with Mn$^{III}$ precursor for the catalytic cycle. Based on the absolute configuration of recovered indolines, (*R*)-**3a** should be oxidized more preferentially than (*S*)-**3a**.

To elucidate the origin of the high level of chiral recognition of azide rac-**1a**, density functional theory (DFT) calculations were performed for the stereo-determining HAT process (Fig. 7). The Gibbs free energies of corresponding transition states follow the spin ordering of triplet < quintet < singlet, and the triplet state was determined to be the ground state (see Table S1 in the Supporting Information). $^3TS_R$ is 1.8 kcal/mol more favorable than $^3TS_S$, which is consistent with experimentally observed stereoselectivity. The effective chiral recognition arises from additional CH···F hydrogen bonding interaction between *tert*-butyl group of (*R*)-**1a** and 2,4-difluorophenyl moiety of catalyst **C12** in $^3TS_R$. This CH···F hydrogen bonding interaction is further confirmed by independent gradient model analysis[64].

In this work, the KR of organic azides through site- and enantioselective C(sp$^3$)−H oxidation is described. The practical manganese catalyzed reaction exhibits good functional group tolerance, and is applicable to a variety of cyclic benzylic azides bearing pharmacologically significant nitrogen-containing heterocycle skeletons with extremely efficient site- and enantio-discrimination. The usefulness of products has also been demonstrated in synthetic applications. Detailed computational studies elucidate the origins of effective chiral recognition involving a hydrogen bonding interaction between substrate and catalyst. This strategically different approach would unlock opportunities for topologically straightforward synthetic planning for KR reactions relying on the reactivity of C(sp$^3$)−H bonds.

## Methods

**General procedure**. To a solution of rac-**1a** (0.1 mmol, 1.0 equiv) in ethyl acetate (1.0 ml) was added **C12** (0.005 mmol, 0.05 equiv) at room temperature. Then PhIO (0.08 mmol, 0.8 equiv) was added as eight portions in 30-min intervals over 3.5 h. After that, the solvent was removed under vacuum and the residue was purified by flash chromatography on silica gel using ethyl acetate/petroleum ether as eluent to give the product (*S*)-**1a**.

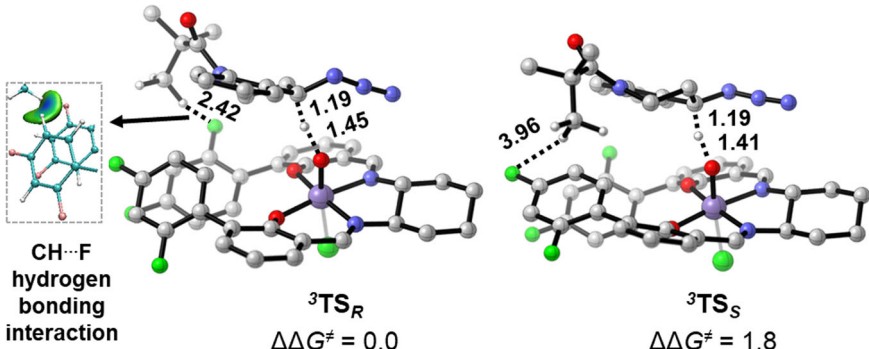

**Fig. 6 Proposed reaction mechanism.** The possible reaction pathway based on our studies and the previous literatures.

**Fig. 7 Geometries and the relative Gibbs free energies of stereoselectivity-determining transition states.** Trivial hydrogen atoms are omitted for clarity. The isosurface of IGM analysis is 0.005. The bond distances are given in Å. All energies are given in kcal/mol.

## Data availability

The authors declare that the data supporting the findings of this study are available within the article and its Supplementary Information files. Extra data are available from the corresponding author upon request. The X-ray crystallographic coordinates for structures reported in Supplementary Information have been deposited at the Cambridge Crystallographic Data Center (**S1**: CCDC 2009823, **S2**: CCDC 2009831). These data could be obtained free of charge from The Cambridge Crystallographic Data Center via www.ccdc.cam.ac.uk/data_request/cif.

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

## Acknowledgements

We gratefully acknowledge the National Science Foundation of China (92156008, 22161142016, 21971148) and Youth Interdiscipline Innovative Research Group of Shandong University (2020QNQT009). The scientific calculations in this paper have been done on the HPC Cloud Platform of Shandong University.

## Author contributions

P.Y. conducted the asymmetric oxygenation experiments and mechanistic studies; A.F. and R.Z. performed the DFT calculations; L.W. prepared the substrates; M.C. initially developed the reaction; L.L. designed the experiments and wrote the paper.

## Competing interests

The authors declare no competing interests.
