## [Peer Review File · Nature Communications]

REVIEWER COMMENTS

Reviewer #1 (Remarks to the Author):

The kinetic resolution of azides is a really useful processes for industrial pharmaceutical and agrochemical development and synthesis. This manuscript reports a new catalytic approach to do this in a fashion that compliments existing approaches. Albeit a few more citations to CuAAC desymmetrisation and kinetic resolution should be included and discussed.

It's extremely well written from the perspective of providing compelling evidence for the hypotheses and the applications of the approach.

The extremely comprehensive supporting material is good - there are three changes required - NMR collection parameters are absent - please add sweep width, temp, number of scans, etc to each spectrum ideally. Secondly, for ee determination by chiral chromatography it would be better to include some representative traces over the whole time course T = 0 onwards - to demonstrate purity as well as ee. Thirdly, and MOST importantly, it says in the ESI that absolute stereochem was determined by XRD, but I could not easily see the Flack parameter - the Flack parameter is REQUIRED to assign absolute (not relative) stereochem - please add Flack parameter to the XRD figure legends and text in the ESI.

there are a few more minor typos in the ESI that should be corrected - please check carefully

In the main text the figure legends must be improved - the legend should, describe in words what we see in the images - currently they do not.

The absence of a non-linear effect does not only imply a monomeric catalyst it could also imply homochiral dimers / oligomers are responsible - what it does show is there is no preference or pathway blocked or accelerated by heterochiral agglomerates - so sharpen up the accuracy of this description a bit please. Not only mono- rather not involving hetero di- etc

One assumes the Kagan formula for determining *s* is used (some great selectivity factors achieved by the way) - please give some details or point to how *s* was determined - citing the protocol or describe a bit more.

Over all exciting and interesting work

Reviewer #2 (Remarks to the Author):

Liu and co-workers have developed a manganese-catalyzed oxidative kinetic resolution for the synthesis of chiral azides. Chiral Mn(salen) complexes were used for the stereoselective oxidation and high selectivity factors were obtained. In the mechanism studies, preliminary KIE studies showed the C-H bond cleavage might be involved in the rate-determining step. DFT calculation of the stereoselectivity-determining TS proposed a CH-F hydrogen bonding interaction. Oxidative kinetic resolution is a well-developed method and the authors also have important contributions in this area (Ref. 48, 50). Chiral azides reported in this work is fairly limited and the other half (oxidative products) are not useful compounds. Below is a list of suggestions/errors that should be considered by the authors.

- (1) How the selectivity factors were calculated should be included.
- (2) In Fig. 2, both 'R' groups of 1q and 1r are 'Me'.
- (3) The purity of compound 6 and 8 in Fig. 4 should be indicated with 'dr' instead of 'ee', as they are diastereomers.

- (4) What's the result with free tetrahydroquinoline or indoline-based azides without nitrogen atom protection?**
- (5) Compound 7 in Fig. 4 is 'estrone derivative' instead of 'estrone'. Related descriptions in main text and also SI should be corrected.**
- (6) Ref. 10 and 12 are repeated.**
- (7) The author proposed a CH-F hydrogen bonding interaction in TS. Is it possible to prove this interaction experimentally?**
- (8) For tetrahydroquinoline or indoline-based azides, different N-acyl groups were utilized. How stereoselectivity was influenced by the size of these protecting groups?**

Reviewer #3 (Remarks to the Author):

In this manuscript, the authors reported a site- and enantioselective sp³ C-H oxidation of racemic cyclic benzylic azides to give highly optically enriched organic azides employing salen manganese complex as chiral catalyst. The reaction proceeded under ambient temperature, and provided good yields (given that the process is a chiral recognition with a maximal theoretical yield of 50%) in a relatively broad substrate scope. The authors demonstrated the scalability of the method to 0.5 mmol scale. Selective oxidation of C-H bonds represents a powerful tool and hot topic in modern organic chemistry, but the studies on enantioselective sp³ C-H oxidation remain very rare. As the manuscript stated, the well-explored oxidative kinetic resolution of benzylic alcohols is the most related precedents. To the best of my knowledge, this work represents the first example of a chiral resolution of organic azides via C-H oxidation. The referee believes that the work would strongly attract the interest of numerous chemists engaged in oxidation and inspire the development of the hot area in modern organic chemistry. Based on the above reasons, the referee recommends the manuscript to be published in Nature Communications.

Other comments:

- (1) A series of cyclic benzylic azides were demonstrated to be well tolerated with the reaction. I am curious what would be the result for acyclic benzylic azides? What's the reactivity and chiral recognition? If possible, the authors might provide such information at least in the Supporting Information.**
- (2) In fig 2 and 3, the authors demonstrated the scalability to 0.5 mmol scale. What's the result for larger scale, e.g. gram scale?**
- (3) While this work focuses on the chiral recognition of organic azides, the representative examples on catalytic asymmetric synthesis of organic azides through other strategies should be included.**

As instructed, we have attempted to succinctly explain changes made in reaction to all comments. Here we reply to each comment in point-by-point fashion.

Reviewer 1 comment:

Comment:

The kinetic resolution of azides is a really useful processes for industrial pharmaceutical and agrochemical development and synthesis. This manuscript reports a new catalytic approach to do this in a fashion that compliments existing approaches. Albeit a few more citations to CuAAC desymmetrisation and kinetic resolution should be included and discussed. It's extremely well written from the perspective of providing compelling evidence for the hypotheses and the applications of the approach.

Response:

Thanks for the reviewer's comment. We previously cited precedents regarding kinetic resolution of organic azides to prepare optically pure azides through azide-alkyne cycloaddition (AAC) in ref. 22 and 23. As suggested by the reviewer, several representative precedents regarding CuAAC desymmetrization and kinetic resolution have been included in the revised manuscript as references 24-26, though these methods cannot provide optically pure azides.

Comment:

The extremely comprehensive supporting material is good - there are three changes required - NMR collection parameters are absent - please add sweep width, temp, number of scans, etc to each spectrum ideally. Secondly, for ee determination by chiral chromatography it would be better to include some representative traces over the whole time course $T = 0$ onwards - to demonstrate purity as well as ee. Thirdly, and MOST importantly, it says in the ESI that absolute stereochem was determined by XRD, but I could not easily see the Flack parameter - the Flack parameter is REQUIRED to assign absolute (not relative) stereochem - please add Flack parameter to the XRD figure legends and text in the ESI.

Response:

Thanks for the reviewer's comments. As suggested by the reviewer, all three changes have been made in the revised Supplementary Information. (1) The general

information of sweep width, temperature, and number of scans has been included in General Information section of the revised Supplementary Information, unless otherwise noted in the section of Analytical Data for Products. (2) Some representative traces over the whole time course $T = 0$ onwards have been included in the revised Supplementary Information. (3) The Flack parameters for both tetrahydroquinoline (flack parameter is 0.10(16)) and indoline-based azides (flack parameter is 0.3(2)) have been added to the XRD figure legends and text in the revised Supplementary Information. The flack parameter for indoline-based azide is somewhat large. We tried to develop the crystal. The XRD exhibited the same configuration, but the flack parameter was still around 0.3. To further confirm the absolute configuration of indoline-based azides, CD spectroscopy measurements and theoretical calculations are employed. The related data has been included in the revised Supplementary Information.

Comment:

There are a few more minor typos in the ESI that should be corrected - please check carefully.

Response:

Thanks for the reviewer's careful proofreading. We have checked and revised the Supplementary Information carefully.

Comment:

In the main text, the figure legends must be improved - the legend should, describe in words what we see in the images - currently they do not.

Response:

As suggested by the reviewer, the figure legends in the main text have been improved, and detailed description in words have been included.

Comment:

The absence of a non-linear effect does not only imply a monomeric catalyst it could also imply homochiral dimers / oligomers are responsible - what it does show is there is no preference or pathway blocked or accelerated by heterochiral agglomerates - so sharpen up the accuracy of this description a bit please. Not only mono- rather not involving hetero di- etc.

Response:

We appreciate for the reviewer's insightful opinion. As suggested by the reviewer, the description has been revised as "The absence of nonlinear effects indicated that the reaction might not involve heterochiral agglomerates" in the main text.

Comment:

One assumes the Kagan formula for determining s is used (some great selectivity

factors achieved by the way) - please give some details or point to how *s* was determined - citing the protocol or describe a bit more.

Response:

As suggested by the reviewer, the method for determining *s* value has been included in the footnote d of Table 1 as ^dSelectivity (*s*) values were calculated through the equation $s = \ln[(1 - C)(1 - ee)]/\ln[(1 - C)(1 + ee)]$, where *C* is the conversion.

Comment:

Over all exciting and interesting work

Response:

We appreciate for the reviewer's comment, and we are glad to know the reviewer like our work.

Reviewer 2 comment:

Comment:

Liu and co-workers have developed a manganese-catalyzed oxidative kinetic resolution for the synthesis of chiral azides. Chiral Mn(salen) complexes were used for the stereoselective oxidation and high selectivity factors were obtained. In the mechanism studies, preliminary KIE studies showed the C-H bond cleavage might be involved in the rate-determining step. DFT calculation of the stereoselectivity-determining TS proposed a CH-F hydrogen bonding interaction. Oxidative kinetic resolution is a well-developed method and the authors also have important contributions in this area (Ref. 48, 50). Chiral azides reported in this work is fairly limited and the other half (oxidative products) are not useful compounds.

Response:

We appreciated for the reviewer's comment. At the end of the comment, the reviewer said that chiral azides reported in this work is fairly limited. We admit that the method is mainly suitable for cyclic benzylic azides including a range of tetrahydroquinoline- and indoline-based substrates with excellent site- and enantio-discrimination. Encouraged by the reviewer's comment, we further explored the applicability of the method in kinetic resolution of acyclic organic azides (Scheme R1). Under the standard conditions, oxidative kinetic resolution of acyclic rac-**A** proceeded smoothly

Scheme R1. Oxidative kinetic resolution of acyclic organic azide.

at rt, and after 12 h the enantiomer was recovered in 51% yield with 48% ee ($s = 4.7$). The result has been included in the revised Supplementary Information. We would like to thank the reviewer again for the valuable comment, which prompted us to further explore the scope of the method.

The reviewer also said that the other half (oxidative products) are not useful compounds. As we known, the oxidized products 2,3-dihydroquinolin-4(1*H*)-ones after hydrolysis for THQ-based azides and 2-(acylamino)benzonitriles for indoline-based azides are key subunits or important precursors for biologically significant molecule, though the chirality was not present in these compounds. For example, 2,3-dihydroquinolin-4(1*H*)-one derivatives have been identified with high binding affinities and good selectivities for 5-HT₆ receptor (*Bioorg. Med. Chem. Lett.* 2011, 21, 698); 2-(acylamino)benzonitriles are important precursors for the synthesis of quinazolines (*Eur. J. Org. Chem.* 2016, 4269; *Synthesis* 2015, 1623). We hope the reviewer could agree with us.

Comment:

Below is a list of suggestions/errors that should be considered by the authors.

- (1) How the selectivity factors were calculated should be included.
- (2) In Fig. 2, both ‘R’ groups of **1q** and **1r** are ‘Me’.
- (3) The purity of compound **6** and **8** in Fig. 4 should be indicated with ‘dr’ instead of ‘ee’, as they are diastereomers.

Response:

Thanks for the reviewer’s comments.

- (1) As suggested by the reviewer, the method for determining s value has been included in the footnote d of Table 1 as ^dSelectivity (s) values were calculated through the equation $s = \ln[(1 - C)(1 - ee)]/\ln[(1 - C)(1 + ee)]$, where C is the conversion.
- (2) The R group of **1r** should be Et, which has been corrected.
- (3) The purity of compound **6** and **8** in Fig. 4 have been indicated with ‘dr’ instead of ‘ee’.

Comment:

- (4) What’s the result with free tetrahydroquinoline or indoline-based azides without

nitrogen atom protection?

(5) Compound 7 in Fig. 4 is 'estrone derivative' instead of 'estrone'. Related descriptions in main text and also SI should be corrected.

(6) Ref. 10 and 12 are repeated.

Response:

(4) The free tetrahydroquinoline or indoline-based azides are not compatible with PhIO, and severe background reactions were observed.

(5) For the description of compound 7, 'estrone derivative' has been used instead of 'estrone' in both main text and Supplementary Information.

(6) The repeated ref. 12 has been removed from the manuscript.

Comment:

(8) For tetrahydroquinoline or indoline-based azides, different N-acyl groups were utilized. How stereoselectivity was influenced by the size of these protecting groups?

Response:

We appreciate for the reviewer's suggestion. The reviewer's comment encouraged us to perform more control experiments, which help us further understand the reaction details. We first explored the substituent effect of different acyl groups on tetrahydroquinoline-based azides as shown in Scheme R2 (Fig. 5D in the revised manuscript). Several aspects of the data merit further comment. Firstly, no reaction was observed for *N*-acyl substituted **9a**. Secondly, the oxidized reactivity was gradually enhanced as the increasing numbers of methyl groups at α -position of the carbonyl moiety (**9a-9c** and **1a**). Thirdly, the oxidative reactivity was lost when placing an oxygen atom between the carbonyl and ^tBu groups. In general, more sterically hindered substrates should exhibit lower reactivity than that of less sterically hindered ones. We speculated that the opposite trend observed for THQ-based substrates in Scheme R2 might originate from the non-covalent interaction between the α -alkyl group of carbonyl moiety and Mn(salen) catalyst.

Scheme R2. The *N*-acyl substituent effect for THQ-based azides.

Scheme R3. The *N*-acyl substituent effect for indoline-based azides.

We next explored the substituent effect of different acyl groups on indoline-based azides as shown in Scheme R3 (Fig. 3 and Fig. 5E in the revised manuscript). When R group of the acyl moiety in *rac*-**3** is Me, Et, or ⁿPr, comparable reactivity and selectivity are observed. When R group is ^tBu, no conversion was observed. No oxidative conversion was also observed for deuterated [D]-**3a**, indicating that the sp³ C–H bond at α -position of carbonyl motif is crucial to the reactivity of indoline substrates.

Comment:

(7) The author proposed a CH-F hydrogen bonding interaction in TS. Is it possible to prove this interaction experimentally?

Response:

We appreciate for the reviewer's suggestion. The reviewer's comment encouraged us to perform more control experiments, which help us further understand the reaction details.

(a) We first explored the substituent effect of different acyl groups on tetrahydroquinoline-based azides as shown in Scheme R2 (Fig. 5D in the revised manuscript). Several aspects of the data merit further comment. Firstly, no reaction was observed for *N*-acyl substituted **9a**. Secondly, the oxidized reactivity was gradually enhanced as the increasing numbers of methyl groups at α -position of the carbonyl moiety (**9a-9c** and **1a**). Thirdly, the oxidative reactivity was lost when placing an oxygen atom between the carbonyl and ^tBu groups. In general, more sterically hindered substrates should exhibit lower reactivity than that of less sterically hindered ones. **We speculated that the opposite trend observed for THQ-based substrates in Scheme R2 might originate from the non-covalent interaction between the α -alkyl group of carbonyl moiety and Mn(salen) catalyst.**

Scheme R2. The *N*-acyl substituent effect for THQ-based azides.

Scheme R3. The *N*-acyl substituent effect for indoline-based azides.

(b) We next explored the substituent effect of different acyl groups on indoline-based azides as shown in Scheme R3 (Fig. 3 and Fig. 5E in the revised manuscript). When R group of the acyl moiety in *rac*-**3** is Me, Et, or ⁿPr, comparable reactivity and selectivity are observed. When R group is ^tBu, no conversion was observed. **No oxidative conversion was also observed for deuterated [D]-3a, indicating that the sp³ C–H bond at α-position of carbonyl motif is crucial to the reactivity of indoline substrates.**

According to the observation in Scheme R2, we postulated that the sp³ C–H bond at β-position of the carbonyl motif is crucial to the reactivity of THQ substrates. According to the observation in Scheme R3, we postulated that the sp³ C–H bond at α-position of the carbonyl motif is crucial to the reactivity of indoline substrates.

(c) We then investigated the substituent effect at 2'-position of 1,1'-biphenyl of the chiral catalyst shown in Scheme R4 (Table 1 in the revised manuscript). When the R group in the catalyst is H (**C8**), *s* value of 2.8 was obtained. Replacing the H group with both electron-donating MeO group (**C9**, *s* = 5.1) and electron-withdrawing Cl and F (**C10**, *s* = 9.5; **C11**, *s* = 25) afforded an enhanced selectivity, suggesting that the selectivity might not originate from CH⋯π interaction between *tert*-butyl group of (*R*)-**1a** and 1'-aryl moiety of chiral catalyst **C**. To our knowledge, three substituents in **C9-C11** including MeO, Cl, and F can effectively form hydrogen bonding with suitable C–H bond. According, the calculated CH⋯F hydrogen bonding interaction in ³**TS_R** (Fig. 7) is reasonable.

Scheme R4. The substituent effect at 2'-position of 1,1'-biphenyl of the chiral catalyst.

Reviewer 3 comment:

Comment:

In this manuscript, the authors reported a site- and enantioselective sp^3 C-H oxidation of racemic cyclic benzylic azides to give highly optically enriched organic azides employing salen manganese complex as chiral catalyst. The reaction proceeded under ambient temperature, and provided good yields (given that the process is a chiral recognition with a maximal theoretical yield of 50%) in a relatively broad substrate scope. The authors demonstrated the scalability of the method to 0.5 mmol scale. Selective oxidation of C-H bonds represents a powerful tool and hot topic in modern organic chemistry, but the studies on enantioselective sp^3 C-H oxidation remain very rare. As the manuscript stated, the well-explored oxidative kinetic resolution of benzylic alcohols is the most related precedents. To the best of my knowledge, this work represents the first example of a chiral resolution of organic azides via C-H oxidation. The referee believes that the work would strongly attract the interest of numerous chemists engaged in oxidation and inspire the development of the hot area in modern organic chemistry. Based on the above reasons, the referee recommends the manuscript to be published in Nature Communications.

Response:

We appreciate for the reviewer's comment, and we are glad to know the reviewer like our work.

Comment:

A series of cyclic benzylic azides were demonstrated to be well tolerated with the reaction. I am curious what would be the result for acyclic benzylic azides? What's the reactivity and chiral recognition? If possible, the authors might provide such information at least in the Supporting Information.

Response:

We appreciated for the reviewer's comment. Encouraged by the reviewer's comment, we further explored the applicability of the method in kinetic resolution of acyclic organic azides (Scheme R1). Under the standard conditions, oxidative kinetic resolution of acyclic rac-**A** proceeded smoothly at rt, and after 12 h the enantiomer

Scheme R1. Oxidative kinetic resolution of acyclic organic azide.

was recovered in 51% yield with 48% ee ($s = 4.7$). The result has been included in the revised Supplementary Information. We would like to thank the reviewer again for the valuable comment, which prompted us to further explore the scope of the method.

Comment:

In fig 2 and 3, the authors demonstrated the scalability to 0.5 mmol scale. What's the result for larger scale, e.g. gram scale?

Response:

As suggested by the reviewer, the gram scale reaction of rac-1a was performed. Under the standard conditions, a comparable result was observed ($s = 32$, 1.0 gram scale versus $s = 43$, 0.5 mmol scale), suggesting that the reaction is suitable for gram scale preparation.

Comment:

While this work focuses on the chiral recognition of organic azides, the representative examples on catalytic asymmetric synthesis of organic azides through other strategies should be included.

Response:

We appreciate for the reviewer's suggestion. As suggested, several representative examples on catalytic asymmetric synthesis of organic azides through other strategies have been included as ref. 18-21.

We thank you for your constructive comments and feedback. We tried our best to address each of your points in detail. We feel the quality of the revised manuscript is much improved and hope you agree.

Thank you for your efforts on behalf of this manuscript.

With sincere regards,

Lei Liu,

Shandong University
Jinan 250100, P. R. China
Email: leiliu@sdu.edu.cn

REVIEWERS' COMMENTS

Reviewer #1 (Remarks to the Author):

the revisions have been made in line with suggestions from the referees - the manuscript is suitable for publication

Reviewer #2 (Remarks to the Author):

The kinetic resolution of azide in this work is less interesting due to the 'specialized' substrate. The most novel aspect of this work is the identification and proposed interaction of the catalyst C-F group with benzylic H of the substrate. The authors have provided more indirect studies on the role of the CF group. If the authors are able to provide further direct studies ie ^{19}F NMR particularly 2D NMR and/or deuteration studies, it will make this work more impactful.

Reviewer #3 (Remarks to the Author):

The author has carefully revised the manuscript according to the requirements and comments of the reviewers, I think the manuscript has met the publishing requirements, and it is recommended to be accepted.

As instructed, we have attempted to succinctly explain changes made in reaction to all comments. Here we reply to each comment in point-by-point fashion.

Reviewer 2 comment:

Comment:

The kinetic resolution of azide in this work is less interesting due to the 'specialized' substrate.

Response:

Thanks for the reviewer's comment. We admit that the method in this work is mainly suitable for cyclic benzylic azides including a range of tetrahydroquinoline- and indoline-based substrates with excellent site- and enantio-discrimination. Moreover, during the first round revision, we further demonstrated that acyclic benzylic azides were also suitable substrates for the method, though a modest selectivity factor was observed ($s = 4.7$, see page S28 in the Supplementary Information). Given the facts that current catalytic kinetic resolution methods dominantly rely on the manipulation of reactive functional groups, and that catalytic kinetic resolution of organic azides remains a formidable challenge due to the small size and instability of the azido group, the method in this work provides a proof of concept of designing a kinetic resolution reaction of challenging substrates through site- and enantioselective C(sp³)-H oxidation. We envisioned that this strategically different approach would unlock opportunities for topologically straightforward synthetic planning for kinetic resolution reactions relying on the reactivity of C(sp³) - H bonds.

Comment:

The most novel aspect of this work is the identification and proposed interaction of the catalyst C-F group with benzylic H of the substrate. The authors have provided more indirect studies on the role of the CF group. If the authors are able to provide further direct studies ie ¹⁹F NMR particularly 2D NMR and/or deuteration studies, it will make this work more impactful.

Response:

We appreciate for the reviewer's suggestion. Firstly, the proposed CH...F hydrogen bonding interaction should be between *tert*-butyl group of substrate (*R*)-**1a** and 2,4-difluorophenyl moiety of catalyst **C12**, but not benzylic hydrogen of (*R*)-**1a** and

catalyst **C12**. Secondly, we envision that the most novel aspect of this work should be the proof of concept of designing a kinetic resolution reaction of challenging substrates through site- and enantioselective C(sp³)-H oxidation, which would unlock opportunities for topologically straightforward synthetic planning for kinetic resolution reactions relying on the reactivity of C(sp³) - H bonds.

Encouraged by the reviewer's suggestion, we tried to perform the NMR for the chiral manganese catalyst. Probably due to the paramagnetic property of the Mn(salen) complex, we did not obtain effective information for ¹⁹F NMR and 2D NOE NMR. Due to the inaccessibility to deuterated pivaloyl chloride derivative, we cannot prepare deuterated THQ substrate at present. Instead, during the first round revision, the deuteration effect of the *N*-acyl moiety of indoline-based substrates had been evaluated (see Fig. 5E). No oxidative conversion was observed for [D]-**3a**, indicating that the sp³ C-H bond at α -position of carbonyl motif is crucial to the reactivity of substrate **3a**. The observation supports the hydrogen bonding interaction.

We thank you for your constructive comments and feedback. We tried our best to address each of your points in detail. We feel the quality of the revised manuscript is much improved and hope you agree.

Thank you for your efforts on behalf of this manuscript.

With sincere regards,

Lei Liu,
Shandong University
Jinan 250100, P. R. China
Email: leiliu@sdu.edu.cn